# Research on Financial Poverty Alleviation Aid for Increasing the Incomes of Low-Income Chinese Farmers

Huibo Pan, Lili Yao *, Chenhe Zhang, Yuchi Zhang and Yuying Gao

School of Economics & Management, Northwest Agriculture and Forestry University, Yangling 712100, China; phb@nwafu.edu.cn (H.P.); zch34@nwafu.edu.cn (C.Z.); 2020010044@nwafu.edu.cn (Y.Z.); 2493367441@nwafu.edu.cn (Y.G.)
* Correspondence: liliyao@nwafu.edu.cn

**Abstract:** Unlike the definition of absolute poverty in international society, rural poverty in China is characterized by farmers' low ability to increase their income, and the unsustainability of income increases. This study examines farming households' issues with increasing their incomes via financial aid from the International Fund for Agricultural Development (IFAD). Through quantile regression and stepwise regression models, this paper studies two ways in which IFAD loan projects can affect farmers' income: by directly promoting increases in farmers' income and by indirectly promoting increases in farmers' income through newly operational agricultural entities. This paper uses the entropy weight and comprehensive evaluation methods to construct an IFAD evaluation index system to evaluate the endogenous development ability of farmers participating in IFAD loan projects. Our empirical results show that IFAD projects significantly and positively affect farmers' income. Our heterogeneity analysis shows that IFAD projects have varied effects on farmers' income growth at different income levels; the lower the income level, the more pronounced the promotion. IFAD loan projects promote farmers' incomes through newly operational agricultural entities, the mechanism of which is their ESG performances. On average, the anti-poverty masses and areas participating in IFAD loan projects show a robust endogenous development impetus.

**Keywords:** increase in farmers' income; International Fund for Agricultural Development; new agricultural operation subject; ESG performance; quantile regression method; poverty alleviation





## 1. Introduction

Since 2012, China has placed significant emphasis on boosting the incomes of rural households, resulting in rapid growth and notable progress in poverty alleviation [1]. Firstly, according to China's practice, financial support is one of the essential drivers of development for people and regions to be lifted out of poverty. Sustainable economic and social development in poverty-stricken areas and stability in poverty alleviation are achievements that cannot be realized without financial support [2]. Secondly, to implement its rural revitalization strategy and achieve the goal of shared prosperity, China should carefully allocate resources to rural financial factors. New types of agricultural operating bodies are helpful in promoting the development of rural industry, so it is crucial to enable credit transactions and support them. However, due to the lack of demand efficiency, it is not easy to increase the balanced output with the expansion of financial function and the increment of credit. The financial literacy of these new rural operators is an evident constraint [3].

The Chinese government announced that the country had been comprehensively lifted out of poverty in November 2021. Still, the people and areas that were lifted out of poverty are weak links in the country's comprehensive rural revitalization and shared prosperity. The current stage's main task is to stimulate the endogenous motivation of these people and areas for further poverty alleviation. For instance, the former impoverished areas targeted in the poverty relief strategy have been lifted out of poverty; however, their living

conditions, productive skills, and infrastructure in various aspects are still fragile, and the possibility of falling into poverty due to illnesses, disasters, or industrial failure is high. The lack of endogenous motivation to alleviate poverty will lead to the formation of a culture of poverty, the intergenerational spread of poverty, and further difficulties in poverty alleviation. The link between poverty alleviation and rural revitalization needs further attention. The governance system of targeted poverty alleviation policies aimed at "enhancing endogenous dynamics" has had an essential impact on the people and areas lifted out of poverty, and in the long run, empowering people to lift themselves out of poverty is conducive to eradicating rural poverty [4].

This paper focuses on how disadvantaged rural households can increase their incomes with the International Fund for Agricultural Development (IFAD). The IFAD, an international financial organization specializing in global poverty alleviation and agricultural development, provides loans with concessional terms to members of developing countries. The IFAD supports member countries in promoting food production, improving the living standards of those who are disadvantaged in rural areas, and eradicating rural poverty. Based on the implementation of its business plan model, this project innovatively uses project industry funds to build interdependent value chains by enabling project funds for enterprises and providing point-to-point support for farmers. This not only ensures that project funds can be accessed by a vast number of vulnerable small-sized farming households but also facilitates the integration of small-sized farmers into the local agricultural value chain, aiding enterprises in building a complete value chain.To a large extent, the project aims to alleviate the agricultural industry's "employment difficulties" [5–8].

In China's dualistic social structure, this study notes socioeconomically disadvantaged groups. It focuses on their learning effects from the perspective of the trickle-down theory with quantile regression modeling, stepwise regression modeling, entropy weighting, and comprehensive evaluation methods. Scholars have concluded that financial support that provides financial subsidies reduces people's objective endogenous motivation to lift themselves out of poverty; however, actively supporting their employment and entrepreneurship can significantly increase their subjective endogenous motivation to lift themselves out of poverty and increase their incomes [9]. In contrast, studies have found that the effect of financial poverty alleviation on farmers' income in China is insignificant [10]. This paper aims to investigate the impact of financial aid on the income trajectories of low-income farmers, and it provides a more detailed analysis of financial aid policies for disadvantaged farmers to fill the gap in the existing literature.

## 2. Literature Review

Solving the issue of rural underdevelopment and increasing farmers' incomes are meaningful steps toward achieving agricultural modernization and ensuring the steady progress of China's national economy [11]. Financial inclusion and agricultural assistance programs are issues of importance and increasing concern worldwide, mainly to policymakers across Africa and the rest of the developing world [12,13]. A financially inclusive society leads to more substantial growth. Financial inclusion aims at providing accessible and affordable access to financial products and services. The main concern for any developing nation from a growth point of view is the advancement of low-income rural populations in addition to high-income populations. Toward this aim, identifying the key determinants that would lead to the successful financial inclusion of low-income rural populations is an important step [14]. Previous studies have demonstrated causal effects between developing countries' governance and the effectiveness of aid. These studies emphasize how good governance and policies impact the effectiveness of aid [15–20]. The IFAD dataset is divided into the following geographical regions: Asia, Africa, Europe, Latin America, and the Caribbean. Kim et al. analyzed the topic in different areas [21,22]. Studies have directly and indirectly explained the importance of the government's role in implementing cooperative development projects. Such studies show that if the governance of the recipient countries is well-organized, there is a high probability that the aid implementation will

be transparent and effective [23]. For other agricultural assistance programs, Totobesola's work highlighted the critical need for a paradigm shift of current research and development programs, as demonstrated by the findings of a project implemented in Burkina Faso, the Democratic Republic of Congo and Uganda by the United Nations Rome-based Agencies (RBAs)—The Food and Agriculture Organization of the United Nations (FAO), the International Fund for Agricultural Development (IFAD) and the World Food Program (WFP) [24]. Among the factors that restrict the development of industry and the increase in farmers' income in rural areas, the investment of credit funds is a vital resource constraint.

This research encompasses two literature streams: the impact of financial poverty alleviation on the increase in farmers' income and the influence of financial poverty alleviation on farmers' income through the establishment of new agricultural operating entities. Regarding the research stream on the impact of financial poverty alleviation on rural households' income increase, Binswanger and Khandker and Shahidur et al. analyzed agricultural loans in India and Bangladesh, respectively, and found that loans had a significant positive impact on farmers' income, agricultural production, and household employment [25,26]. Khandker and Faruqee, taking agricultural loans in Pakistan as an example, pointed out that agricultural loans have a noticeable positive correlation with increasing farmers' income [27]. However, the large number of agricultural loans has disadvantaged policy banks' development, since they need more government subsidies to be sustainable [28]. Poor rural communities in the developing world are the IFAD's principal target groups [29]. On the one hand, financial inclusion is critical, as increasing the disadvantaged's access to financial services is often considered an effective tool that can help reduce poverty and lower income inequality [30]. The Indian government has been working on providing access to financial services to the people living in poverty and the deprived section since early times [14].

Conversely, for low-and-middle-income countries (LMICs), broad development interventions such as financial and digital connectivity and women's empowerment loom more important than narrowly focused interventions regarding progress in the food system [31]. The main body of agricultural business in Africa is dominated by individual farmers, who lack advanced farm technology and the necessary agricultural infrastructure, resulting in a lack of adaptability and resistance in the face of external risks and unstable agricultural harvests, which is a situation that makes the investment environment for agriculture in Africa less than ideal. The development of agricultural financial services can improve the resilience of African agriculture, strengthen the African agricultural production supply chain, and promote the sustainable development of African agriculture [32]. Wang et al. constructed a theoretical model of inclusive finance and financial poverty alleviation by using the classical model of agricultural families, and the authors stated that financial support has a stimulating effect on farmers and an opportunity effect on poor farmers by endowing them with endogenous development impetus through finance [33]. Yang utilizes Heckman's sample selection model, the Oaxaca–Blinder model difference decomposition method, and the redistribution influence function regression-based distribution decomposition method to analyze the impact of digital financial development on workers' employment income, and he claims that with financial support, the disadvantaged people have alleviated poverty and improved the imbalance of income distribution [34]. Jiang et al. analyzed how capital investment promotes the growth of farmers' income by using the labor–capital production function model and the difference in different methods to build the model for empirical analysis [35].

Another literature stream is the impact of financial poverty alleviation on farmers' income growth through new agricultural business entities. Financial support for the coordinated development of new agricultural business entities and small farmers should focus on developing rural industries and allocate more financial resources to critical areas and weak links in rural areas [36]. Financial inclusion provides opportunities for new business creation, growing existing ones, and improving income for entrepreneurs [37]. These business investments would increase macroeconomic factors like GDP and per capita

income, thus benefiting the nation. If not invested in the business, these loans could be utilized for education or infrastructure development purposes, enhancing the standard of living. A financially well-included society is a win–win situation for nations, banks, financial institutions, and, most notably, all households [14]. Balibae's study investigates the impact of inclusive financial development on the performance of inclusive finance for agricultural start-ups in 30 sub-Saharan African countries from 2009 to 2017. The study justifies that inclusive financial development promotes entrepreneurial activity by meeting the needs of firms for accessible, affordable, and broad-based credit and efficient and cost-effective risk management for the benefit of investors and firms [38]. Xue et al. explained how finance affects farmers' income from poverty reduction and income distribution. Those researchers stated that local industry upgrading is an intermediary in financial support to increase farmers' income [39]. Zhou et al. believed that new agricultural entities in financial poverty alleviation can provide poor households with inputs and services throughout the agricultural production process by establishing close cooperation mechanisms with poor households; those entities may also pay dividends or wages to the disadvantaged who have joined the development of new agricultural entities in the form of land equity or labor employment, effectively stimulate the poor population to enhance production and operation capacity and economic efficiency and achieve stable poverty eradication [40]. Xia et al. found that new agricultural operators promote the integration of internal resources and external support, such as financial poverty alleviation, between relatively poor groups and poor areas through the practical input, acceptance, and transfer of resources. It is conducive to improving the effectiveness of poverty alleviation and the vitality of regional development. Under a good social environment, market conditions, and favorable policy support, the poverty reduction effect of enterprises is the most obvious and more conducive to sustained income increase, poverty alleviation, and sustainable development of poor areas [41]. As the ideology of green and sustainable development, the green development behavior and performance of the new agricultural entities determine the development direction of green agriculture in the country [42]; the ESG performance of enterprises can promote green innovation and give enterprises an advantage in market competition [43,44]. Favored ESG performance effectively signals that companies are paying more attention to business compliance and sustainability, which could avoid short-sighted behavior in development as much as possible [45,46].

The literature research reflects the new trend of exploring the impact of precision poverty alleviation on farmers from the household perspective; it aligns with the practice demand of "anti-poverty stability investigation" after poverty alleviation [47]. This paper aims to study the impact of financial aid on low-income farm households' income-increasing paths and assess the effectiveness of the financial aid program. Current studies analyze financial support in rural areas of the country as a whole or the concentrated impact on all industries in a particular region. The research gap is, therefore, for a more detailed analysis of the effects of financial aid poverty alleviation on income generation among rural low-income farmers. In this paper, the regions with specialty industries having financial aid were investigated, and the impact of financial aid on the income increase in rural households was analyzed more thoroughly. The research conclusion could provide insights into China's rural development, encourage rural areas to combine the characteristics of regional industries, and increase financial aid for agricultural business entities to realize the goal of increasing the income of rural low-income farm households through the development of industries and then promote shared prosperity.

This paper utilizes quantile regression and stepwise regression models to examine the two paths of IFAD loan projects affecting the increase in farmers' income in the areas having IFAD aids. Entropy weight and comprehensive evaluation methods are used to construct the evaluation index system of IFAD for the endogenous development capability of poverty-alleviated households. The possible marginal contribution of this paper is as follows: First, by combining the statistical analysis and the empirical method, the paper explores the two paths of financial support promoting poverty-relief people and the income-increasing

effect in the poverty-relief areas from both theoretical and empirical aspects, which could enrich the rural financial development theory. Second, this research establishes indicators for evaluating the endogenous development capacity of IFAD-supported households and regions in poverty alleviation, considering the perspectives of human capital, financial capital, natural capital, and social capital. A synthesis of comprehensive empirical analysis makes the research conclusion more accurate and reliable. Third, this paper incorporates IFAD-supposed new agricultural entities' ESG performance to expand the literature on the combination of ESG performance and farmers' income increase, which could also provide insightful implications for China's rural revitalization and sustainable development. After careful examination, the research ideology could be extended to counterparts of China at similar development stages or with similar endowments. Last, the field survey and interview data used in this paper can reflect the actual situation of agriculture, rural areas, and farmers, which have vital timeliness, representativeness, and practical significance.

## 3. Theoretical Analysis and Research Hypotheses

Fundamentally, this study explores the impact of IFAD's support in China's dualistic social structure. It examines explicitly socioeconomically disadvantaged groups and their learning effects within the trickle-down theory. The objective is to explain how IFAD's assistance contributes to increasing farmers' income and reducing poverty from an economic development perspective. The study seeks to attain sustainable development goals by employing the "first-rich leads after-rich" approach, focusing on employment and consumption. Farmers and agricultural entities participating in IFAD loan projects have been leading in China's rural revitalization and shared prosperity. For farmers, priority is given to relatively disadvantaged households who join the project voluntarily with direct grant transfer for productive inputs, field facilities, and farming equipment. The IFAD–Shaanxi loan program supports the enterprises' intensive production, post-production infrastructure, and equipment investment by giving local disadvantaged households agricultural enterprises quantitative shares or fixed dividends. As the project and the investment of funds gradually formed a "government–elite–farmers" chain, it became more feasible to encourage projects of elite groups to play a leading role [48]. By actively participating in IFAD projects, elite groups can increase their income and stimulate the internal motivation of other farmers to reduce poverty and increase their income. Elite groups play a bridge role in linking small farming households with large markets by centralizing dispersed small farmers and forming economies of scale. The group uses collective economic cooperatives and other means to relieve ordinary farmers' worries and enhance the trust and support of the government and loan projects. Also, the group improves farmers' farming techniques through demonstration and training to reduce the cost of production and marketing, to increase the income of the disadvantaged, and to reduce poverty. The trickle-down effect shows a positive externality. Through radiation, more disadvantaged people are driven out of poverty and to the road of shared prosperity and sustainable development. The IFAD–Shaanxi project aims to stimulate rural households to upgrade their capacities through the benefits of policies, integrate elements of rural development, stimulate other rural households to participate in market-based operations through a demonstration effect, and stimulate the endogenous motivation of poor households to lift themselves out of poverty to achieve joint development.

The learning effect of IFAD loans can explain the difference in income of different households, and even different households have different learning effects given their different knowledge level and learning abilities. In carrying out the strategy of rural revitalization and constructing the modernization of the agricultural industry, the cultivation of professional farmers plays multiple roles in helping farmers accumulate human capital and promoting the development of rural society. For instance, the strategy could provide farmers fair access to skills training opportunities to lay the foundation for lifelong career development. Through skills training, the strategy provides a human capital guarantee for effectively linking up the achievements of poverty alleviation and rural revitalization.

Then, it provides adequate industrial and economic support for rural revitalization. Also, the strategy provides intellectual support for rural families to increase their income by providing rural families with appropriate training in adult continuing education and skills [49]. In the process of IFAD policy implementation, professional farmers actively participate through the experience of learning effects to achieve "first-rich led after-rich" toward a sustainable development path of shared prosperity. The empirical study of the learning effect shows that the skills acquired through learning are lifelong and embody one's ability and quality, which will play a role in the future labor market and improve the expected labor remuneration of workers, contributing to future career transitions [50]. This learning effect is more evident among young people than other groups because they need to prove themselves through a foothold, higher income, and standard of living. According to the field investigation, young farmers generally show a more vital endogenous development motive force. In sum, IFAD can endow the endogenous development impetus of poverty-relieved households and promote the increase in farmers' income with the learning effect.

Therefore, this study proposes a hypothesis to be tested.

**H1.** *IFAD projects can increase farmers' incomes. The increasing effect is heterogeneous to farmers with different income levels; the lower the income level, the more significant the income increase effect.*

In industrial development, the government does not intervene directly but manages production factors through policy guidance to give full play to the endogenous motive force of the economy. On the one hand, in the implementation process of IFAD, the government, as a bridge connecting new agricultural business entities and farmers, does not provide financial subsidies to them but only utilizes the matching funds to build public infrastructure, such as land consolidation and improvement, rural water supply and power supply systems, and the repair of public roads. It promotes the development of unique economic industries but does not target the special farming households and business entities participating in the fund program. On the other hand, the traditional finance departments still need to implement tax incentives for farmers and new agricultural business entities participating in the loan program but have mainly played a supervisory role [51]. The government-approved IFAD loans take the form of material security; for instance, the program distributes free seedlings and fertilizers to farmers, organizes vocational training to help farmers solve problems such as information asymmetry, and explores distribution channels for farmers' products to promote the formation of a stable industrial cooperation model between farmers and business entities. The program can not only drive farmers from the model of self-production and self-marketing to large-scale business development, but it can indirectly promote the acquisition, processing, and sales related to industries with local characteristics, provide more employment opportunities for local people, and endow households with the endogenous development motivation to lift themselves out of poverty further. The IFAD loan project strengthens industries' vitality with local characteristics and market competitiveness, providing a solid umbrella-like protection for sustained and sound industrial development and farmers' income growth.

The IFAD loan has also played an essential role in the rural model of "cellular society". First of all, the IFAD loan works with agricultural operators to promote the development of local distinctive industries, help rural areas absorb idle labor, encourage farmers to return to their hometown and find employment opportunities, provide farmers with endogenous impetus for development and then improve farmers' income, and enhance the local people's sense of security and well-being. Secondly, the IFAD loan has led to the transformation of industrial structure to scale, standardization, branding, and marketization and encouraged new agricultural entities to perform more ESG responsibilities in management. The IFAD loan project could encourage those agricultural enterprises to pay more attention to economic and social benefits and improve their social responsibility performance.

From the "Umbrella Society" represented by the government and the "Cellular Society" represented by the rural folk forces, the core idea of China's "Dual Society" structure theory was suggested [52]. These two subjects are embedded in the social structure to provide resources for developing rural characteristic industries and promote the development of industries. They are not in opposition to each other but concerning each other. The government allocates the resources as "Umbrella" to provide intermediary support for enterprises to reduce transaction costs. At the same time, with the increase in the supply and demand density of the industrial chain, the folk "Cellular" structure has been gradually strengthened and began to play a role in industrial governance [53]. The government, as an external regulating force, uses the IFAD's loan program as well as its resources and power to provide "Umbrella" support for the "Cellular" of the people, to empower people and areas out of poverty, and to promote the development of rural industries and increase farmers' income.

In China's 14th five-year plan and the 2035 long-term vision, the Chinese government has set out the goal of implementing Green Innovation. In practice, it is a trend for new agricultural enterprises to pay attention to environmental and social effects. Improving ESG performance has become an indispensable way for the sustainable development of many emerging industries. ESG is a new enterprise evaluation method to measure environmental, social, and governance factors [54], and the multidimensional indicators of ESG performance also provide a measure of the effectiveness of related policies. According to the theory of sustainable development, the long-term health of a company determines its performance, which is closely related to its ESG performance. Drawing on the theory of Zhou et al., in the past economic development, agricultural entities expanded scale and increased production at the expense of the environment, which caused irreparable harm to the local environment [55]. At a time when more and more attention is being paid to the environment, drawing on the ideas of Chen et al., the main body of agricultural management should transform in the direction of sustainable development without reducing the output in order to meet the call of the state, comply with the development law of green economy, and complete the perfect transformation of the new era [56,57]. A good ESG performance suggests that agricultural enterprises pay attention to corporate sustainable development with long-term social responsibility. Li et al. suggested that a good ESG performance of enterprises can improve the voice of enterprises in the supply chain and reduce financing pressure, thus forming a benign supply–demand relationship between enterprises and customers [58].

According to the above analysis, this paper puts forward the research hypotheses H2a and H2b to be examined.

**H2a.** *The IFAD loan project indirectly increases farmers' income through agricultural enterprises.*

**H2b.** *Agricultural enterprises' ESG performance is the working channel that promotes the farmers' income.*

Zhang et al. found that in implementing financial poverty relief policies, first of all, rural households should be taken into account to enhance their access to financial products, and secondly, how finance can give endogenous impetus to development should be noted. Technical training has increased rural households' awareness of financial poverty relief policies, thereby increasing their trust and support to prevent backward to poverty [59]. Liu stated that financial support has four primary characteristics: predictable income, accessible information, reduced cost, and controllable risk. He stated that it was important to develop financial services around "All-industry", "All-chain", "All-subject", and "All-round" and enhance the endogenous motive force of financial support for rural industry development and increasing farmers' incomes [60]. As for the endogenous driving forces of development in areas where financial empowerment can lift people out of poverty, they actively cultivate

characteristic-led industries, paying more attention to the construction of characteristic brands; new industries and business forms are emerging [61].

The International Fund for Agricultural Development (IFAD) project aims to integrate people experiencing poverty into the industrial chain by linking the government and agricultural enterprises with three forms: order-oriented farming, quantification of stock share, and employment opportunity. This can not only encourage farmers' enthusiasm to participate in production and management activities but also help county-level enterprises from the source of funds to solve the problem of insufficient endogenous motivation and to achieve the goal of rural income improvement. Specifically, with technical assistance, skills training, and access to employment opportunities, poverty relief people can enhance their endogenous development capacity, effectively promoting the accumulation of human capital, physical capital, social capital, and financial capital [62]. Moreover, the IFAD promotes industrial integration, raises agricultural industrialization, constantly increases the scale of agricultural business entities, and further enhances the radiation-driven effects; also, the financial poverty alleviation model stimulates the inner motive force of the poverty alleviation masses and the poverty alleviation areas [63].

Based on the above analysis, this paper proposes a hypothesis to be tested.

**H3.** *Households and areas lifted from poverty through IFAD loan projects have more substantial endogenous development capacities than non-participants.*

Accordingly, this paper establishes an IFAD aid analysis framework on the impact of farmers' income and endogenous development dynamics (see Figure 1).

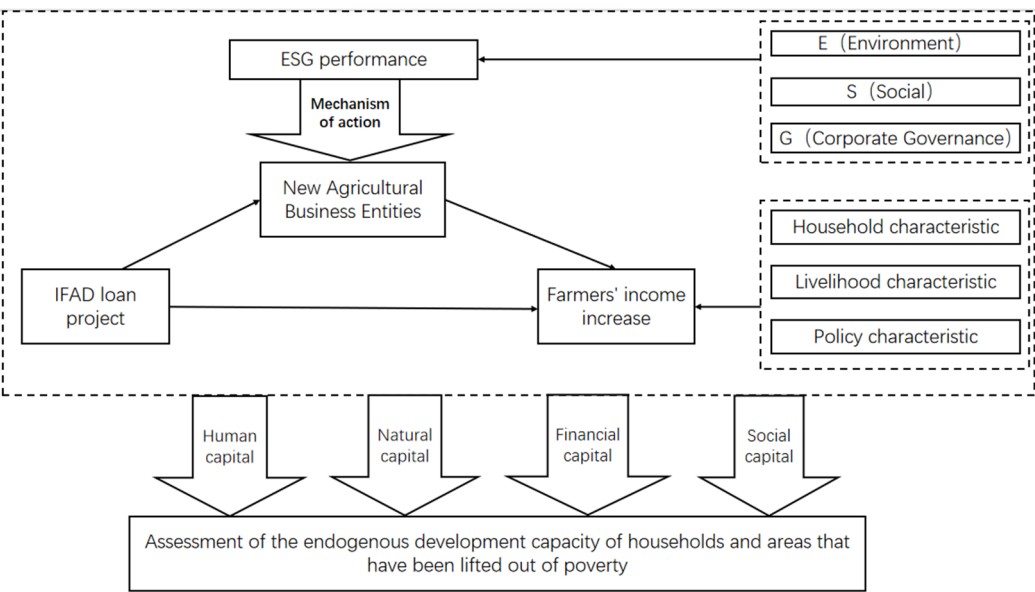

**Figure 1.** The analytical framework for IFAD aid's impact on farm household incomes and endogenous development dynamics.

## 4. Data Sources and Descriptive Statistical Analysis

### 4.1. Data Sources

The IFAD loan project covers Hanzhong, Ankang, and Shangluo counties. For the sample of agricultural business entities, the study obtained information on the business plans of all 358 agricultural business entities participating in the IFAD project through questionnaire surveys and focus group interviews. For the sample of farm households, the team conducted household interviews and questionnaire surveys from major villages in the Nanzheng district. Firstly, the Nanzheng district of Hanzhong City was chosen for empirical analysis for two reasons: more than half of the funded projects were located in

Hanzhong City, and the other two cities (Ankang and Shangluo) suffered natural disasters in the survey year, such as floods, so there may exist exogenous shock. Secondly, the local agricultural bureau provided the demographic data of farmers in each village. Twenty-five villages in Nanzheng district have specialty industries and participate in the IFAD loan program. Thus, five villages, X2–X6, were selected as the treated group with the simple random sampling method according to the principle of 20% population. From the villages with specialty industries but not participating in the project, X1, a village of Nanzheng with a medium level of development was selected as the control group, representing each village's average level. Finally, in each of the selected twenty-five villages, a simple random sampling method was applied to choose 20% of the household population, 110 households in total, for in-depth interviews and surveys [64]. A total of 110 questionnaires were distributed in this survey. In total, 104 out of 110 valid questionnaires were recovered with an effective turnout rate of 94.55%.

*4.2. Concepts Definition*

The dependent variable is the per capita net income of farming households. This paper measured the effectiveness of IFAD financial support in terms of the effect of increasing farm household income (see Table 1).

**Table 1.** Definition and statistical description of variables.

| Variable Settings | | | Statistical Description | |
|---|---|---|---|---|
| | **Variable** | **Definition** | **Mean** | **Standard Deviation** |
| Dependent variable | Per capita net income | Logarithm of per capita net income of farm households | 7.18 | 0.49 |
| Independent variable | Area of agricultural land supported by IFAD | Area of agricultural land supported by IFAD (acres) | 1.36 | 0.67 |
| Control variables    Household head characteristics | Age | <18 years = 1<br>18–30 years = 2<br>31–40 years = 3<br>41–50 years = 4<br>51–60 years = 5<br>61+ years = 6 | 4.83 | 0.83 |
| | Educational attainment | Primary and below = 1<br>Junior high school = 2<br>High school/middle school/technical school = 3<br>University college = 4<br>Undergraduate = 5<br>Master's degree and above = 6 | 2.17 | 1.00 |
| | Physical health status | Very bad = 1<br>Worse = 2<br>General = 3<br>Better = 4<br>Very good = 5 | 3.44 | 1.14 |
| | Population burden ratio | Percentage of population under 16 and over 60 (%) | 0.68 | 0.10 |
| | Number of family farmers | Number of people in the household working in agriculture (persons) | 3.08 | 0.81 |

**Table 1.** *Cont.*

| | | | Variable Settings | | Statistical Description | |
|---|---|---|---|---|---|---|
| | | **Variable** | **Definition** | **Mean** | **Standard Deviation** |
| | | Whether there are village cadres or village representatives | Yes = 1<br>No = 0 | 0.77 | 0.42 |
| | | Received training or not | Yes = 1<br>No = 0 | 0.93 | 0.25 |
| Livelihood characteristic | | Farming | Not farming at all = 1<br>Farming during peak season = 2<br>Full farming = 3 | 2.49 | 0.54 |
| Policy characteristic | | Level of knowledge about IFAD | No knowledge at all = 1<br>No knowledge = 2<br>Understand some = 3<br>Better understanding = 4<br>Very knowledgeable = 5 | 3.48 | 1.25 |

4.2.1. Direct Effects of IFAD Loan Projects on Farmers' Income Growth

The independent variable is the IFAD loan-funded farmland area. This paper focused on the impact of IFAD programs on the income-increasing effects of farm households, so the area of agricultural land aided by IFAD was used as the major explanatory variable.

Control variables: (1) Household characteristics of household head: age, educational attainment, physical health status, farming status, population burden ratio, farming number of family, village cadres or representatives in the household, and occupational training; (2) Livelihood characteristic: family members working in agriculture; and (3) Policy characteristic: level of knowledge about IFAD.

Additionally, for the impact of religion and tradition on the effectiveness of policies to support income development levels, according to Yang et al. and Wu et al., for one thing, the religious beliefs of Chinese farmers include folk religious beliefs and institutional beliefs. The faith base in a region has relative stability and regional differences. For another, the same rural area or village is traditionally homogeneous, with highly similar production and lifestyles dependent on natural endowments, and such traditions are difficult to change [65,66]. The IFAD–Shaanxi loan project for the development of rural specialty industries covers the cities of Hanzhong, Ankang, and Shangluo. The scope of financial support studied in this paper is limited to the southern Shaanxi region; the farmers' ethnic characteristics, belief bases, and traditional ways of life and production are generally the same. Therefore, the influence of religion and tradition on the income level of farmers is limited.

4.2.2. Indirect Driving Effect of IFAD Loan Projects on Farmers' Income Increase through New Agricultural Business Entities

The independent variable is the amount of IFAD project loans.

The mediating variable is the ESG performance of new agricultural enterprises. With Chinese characteristics, China's ESG development has a historic opportunity based on the global advocacy of sustainable development and the strategic background of China's goal of achieving "carbon peak" and "carbon neutrality". China's rural enterprises are responsible for developing the economy, improving people's livelihoods, and revitalizing the countryside; hence, copying Western companies' ESG theories and practical experiences differs from China's national agricultural conditions. Drawing on Xu et al., the embodiment of ESG indicators of Chinese agribusiness are based on the five themes of employee welfare, environmental protection, food safety, social public welfare, and raising farmers to increase

income [67]. Although the social enterprises studied in this paper have yet to achieve certification, they are mainly influenced by financial policies in China's rural revitalization context. Based on reference, this study selected the evaluation standard of the leading ESG indicators by China's current development stage and primary policy guidance. This paper applied the entropy weight method to score the agricultural business entities participating in the IFAD–Shaanxi loan project. It formed the ESG performance of the agricultural business entities and constructed the ESG evaluation indexes with Chinese characteristics. The ESG performance involves environmental, social, and corporate governance. With the requirements of the IFAD loan project, green packaging costs, waste pollution and management policies, and environmental awareness are taken as environmental evaluation indicators, which are mainly measured by the use of green packaging by agricultural business entities and the environmental awareness of enterprises information acquired from interviews; the number of farmers establishing cooperative relationships, the cost of obtaining industrial development loans and product quality and safety are taken as social evaluation indicators, and corporate control and risk management as corporate governance evaluation indicators.

Control variables include the reported amount, payment amount, and total planned investment of new agricultural enterprises. Given the characteristics of the new agricultural business entities and the loan program, unlike listed companies, the beneficiaries of the loan program are primarily small and medium-sized agricultural enterprises in the county. As a grassroots enterprise, local agricultural enterprises' governance structures rely on resource constraints. They pay more attention to their long-term sustainable development in their daily operation process, so the reported amount, payment amount, and total planned investment of different new agricultural business entities under the loan program were chosen as control variables. The explanations of variables and comprehensive evaluation indexes are shown in Tables 2 and 3, respectively.

**Table 2.** Definitions and statistical descriptions of agricultural business variables.

| Variable Settings | | | Statistical Description | |
|---|---|---|---|---|
| **Typology** | **Variable** | **Definition** | **Mean** | **Standard Deviation** |
| Dependent variable | Per capita net income | Logarithm of per capita net income of farm households | 7.18 | 0.49 |
| Independent variable | Amount of IFAD project loans | Amount of IFAD project loans (yuan) | 781,773.724 | 788,430.394 |
| Intermediary variable | ESG performance | ESG evaluation indicator system score | 63.363 | 17.352 |
| Control variables | Amount paid | Amount paid (yuan) | 1,033,230.892 | 1,909,240.197 |
| | Amount reported | Amount reported (yuan) | 662,730.954 | 771,684.761 |
| | Total planned investment | Total planned investment (yuan) | 271,8849.075 | 2,621,541.985 |

### 4.3. Descriptive Statistical Analysis

#### 4.3.1. Descriptive Statistics on Direct Driving Effects

Combining the questionnaire and interview data, in the case of farm households, the average value of IFAD-supported farmland is 1.36 acres, which was about 20% of the cultivated land of the researched farm households. Farmers first observe the income-increasing effect through trial planting on several plots before deciding to plant on a large scale. The farmers, on average, are 4.83 years old with a standard deviation of 0.83, indicating that the majority fall within the 50–60 age range. Interviews revealed that many young laborers in the village opt to work outside, leaving the elderly to manage farm

responsibilities. The average education level is 2.17, which is roughly equivalent to a secondary school education. Those with higher education attainment are advanced in thinking and actively participate in village affairs, leading to country development.

Given that most of the respondents are elderly and in average health, it is unsurprising that the mean physical health status score is 3.44. The mean value of the population burden ratio is 0.68, noting that most households have dependents. Village cadres or village leaders who take part in the IFAD loan program have an average value of 0.77, suggesting village representatives, or cadres, are better qualified to serve as role models probably because they are more intelligent and sophisticated to trying new ideas and trusting of the government. In addition, more than 90% of those involved in the project have received and participated in training. Experts hold regular training sessions for farmers to improve their professional skills. The mean value of farmers' knowledge of IFAD is 3.48. Most farmers only know about it or need help understanding the policy. Therefore, the implication could be that relevant parties should increase publicity to stimulate the endogenous motivation of the people out of poverty, which will help to drive farmers to increase their income through financial support and improve the living standard of the people out of poverty.

**Table 3.** Comprehensive evaluation indicators.

| Target Audience | Content of the Survey | |
| --- | --- | --- |
| | **Dimension** | **Evaluation Indicators** |
| Agricultural business entities | E (Environment) | Green packaging costs ($x_1$) <br> Waste pollution and management policy ($x_2$) <br> Environmental awareness ($x_3$) |
| | S (Social) | Number of cooperative farmers ($y_1$) <br> Cost of obtaining an industrial development loan ($y_2$) <br> Product quality and safety ($y_3$) |
| | G (Corporate Governance) | Corporate governance ($z_1$) <br> Risk management ($z_2$) |

### 4.3.2. Descriptive Statistics on Indirect Driving Effects

Regarding agricultural enterprises, the mean value of its IFAD project loan amount is 781,773.724 yuan, and the standard deviation is 788,430.394. A significant standard deviation indicates that there may be a large gap in the level of development or the development concept among different agricultural business entities. Consequently, a large discrepancy in the number of loan amounts was aided. The mean value of the ESG performance score is 63.363 points, which is in the upper middle range of the scale. In October 2015, China proposed the new development concept of "innovation, coordination, greenness, openness, and sharing". The agricultural entities pay more attention to green development but are still gradually developing. However, studies have shown that agricultural enterprises disclose less information about increasing farmers' income, which may have yet to attract their attention to social responsibility [41]. The mean value of the total planned investment of the new agricultural management subject is 2,718,849.075 yuan, the mean value of the payment amount is 1,033,230.892 yuan, the mean value of the reported amount is 662,730.954 yuan, and the reported ratio is 64%, which is consistent with the 40% self-financing ratio of the agricultural enterprises as stipulated in the project.

## 5. Research Methodology

### 5.1. Quantile Regression Model

In 1978, Roger Koenker and Gilbert Bassett proposed the quantile regression method to minimize the distance between the explanatory variables and the fitted values. Furthermore, quantile regression allows for the observation of the tails of the dependent variable, reflecting more accurately the effect of the independent variable on the shape of the conditional distribution of the dependent variable. Moreover, it does not make any assumptions

about the distribution of the random error term; the results are not easily affected by the extreme value, and the regression is more robust to reflect the data information more comprehensively [68]. To fix the inaccuracy of mean regression estimation and the short-coming of only being able to examine the effect of the covariates on the dependent variable around the mean, scholars usually use quantile regression instead of mean regression. Referring to the literature followed by Zhu et al. [62], in this study, the net incomes of farm households participating in IFAD loan programs were grouped according to quartiles to examine the impact of IFAD on poverty relief households at various income ranges and their heterogeneity. According to the conditional quartile of the dependent variable "per capita net income", the model was set as follows:

$$LnY_{i,d} = \alpha_{0,d} + \alpha_{l,d} loan_{i,d} + \alpha_{k,d} X_{i,d} + \varepsilon_{i,d} \tag{1}$$

where $LnY$ denotes the per capita net income of the households participating in IFAD loans to fight poverty, taking logarithmic values; the subscript $i$ denotes the ith farm household, and the subscript $d$ denotes the quartile, with $d$ = 25%, 50%, and 75%, which denotes a total of three income quartiles of 0.25, 0.50, and 0.75, representing the low-income, middle-income, and high-income populations participating in the IFAD loans poverty relief program. The core explanatory variable, *loan* represents the area of cultivated land supported by IFAD loans, reflecting the implementation of IFAD's loan policy; the control variable, $X$, reflects the characteristics of farm households, livelihood capital, and policy; and $\alpha$ is the semi-elasticity coefficient, which represents the percentage of the change in per capita net income of farm households caused by a one-unit change in the core explanatory variable, *loan*. $\varepsilon$ denotes the random error term. k denotes the number of farmers.

### 5.2. Stepwise Regression Model

Referring to Liu's study, stepwise regression is a method based on linear regression. The idea is to introduce variables one after another and, after introducing a new variable, test the original variables selected for the regression model one by one, removing those not considered meaningful until no new variables are introduced without removing original variables, thus ensuring that every variable in the regression model is meaningful [69]. The stepwise regression method was used to test the relationship among industrial development loan programs, farmers' income increase, and ESG performance of agricultural business entities.

In order to test the direct impact of industrial development loans on farmers' income, the following model was constructed:

$$HI_{it} = \alpha_0 + \alpha_1 IDL_{it-1} + \sum \alpha_k \, Controls_{it} + \varepsilon_{it} \tag{2}$$

In order to test the impact of industrial development loans on the ESG performance of agricultural enterprises, the following model was constructed:

$$ESG_{it} = \beta_0 + \beta_1 IDL_{it-1} + \sum \beta_k \, Controls_{it} + \varepsilon_{it} \tag{3}$$

In order to test the impact of agricultural entitizes' ESG performance on industrial development loans and farmers' income increase, the model was constructed as follows:

$$HI_{it} = \gamma_0 + \gamma_1 IDL_{it-1} + \gamma_2 ESG_{it} + \sum \gamma_k \, Controls_{it} + \varepsilon_{it} \tag{4}$$

In the above three models, $i$ represents the firm; $t$ represents the year; $HI_{it}$ denotes the increase in farm household income of firm $i$ in year $t$; $IDL_{it-1}$ denotes the industrial development loan received by firm $i$ in year $t-1$; $ESG_{it}$ denotes the ESG performance of firm $i$ in year $t$; $Controls_{it}$ denotes the relevant control variables for firm $i$ in year $t$; $\varepsilon_{it}$ is the residual term; $k$ denotes the number of *Controls*; $\alpha$, $\beta$, $\gamma$ are the coefficients.

### 5.3. Entropy Weight Method

The entropy weighting method is used to determine the weights, and entropy can measure the degree of disorder in the system; the smaller an indicator's entropy, the more information the indicator provides. The higher the indicator's degree of variability (variance), the greater the role played in the comprehensive evaluation, and the higher the weight should be. The basic principle of the entropy weighting method is to determine the objective weights according to the magnitude of indicator variability. Referring to the research method of Xie et al. [70], the improved entropy weight method was used to evaluate the information of the main body of agricultural management for ESG and to assess the endogenous development ability of poverty-eradicating households. The method eliminates the influence of positive and negative indicators and null values and avoids the subjective opinion of traditional expert scoring methods in determining weights.

#### 5.3.1. Data Standardization

Positive indicators:

$$x'_{ij} = \frac{x_{ij} - min(x_{ij})}{max(x_{ij}) - min(x_{ij})} + 0.0001 \tag{5}$$

Negative indicators:

$$x'_{ij} = \frac{max(x_{ij}) - x_{ij}}{max(x_{ij}) - min(x_{ij})} + 0.0001 \tag{6}$$

where $x'_{ij}$ is the value after standardized panning of farmer $j$ indicators, $x_{ij}$ is the actual value; $max(x_{ij})$ and $min(x_{ij})$ are the maximum and minimum values of farmer $i$'s $j$ indicators, respectively.

#### 5.3.2. Determination of Indicator Weights

Different indicators affect the endogenous development capacity of farm households to different degrees; it is necessary to determine the weight of each indicator. The study adopted the entropy value method to determine the weights of different indicators considering the characteristics of the data. First, the weight of each farm household on each assessment indicator $p_{ij}$ was calculated:

$$P_{ij} = \frac{x'_{ij}}{\sum_{j=1}^{p} x'_{ij}} \tag{7}$$

Then, the entropy value of the $j$th indicator was calculated. The larger the entropy value, the smaller the influence and the smaller the weight:

$$e_j = -k \sum_{i=1}^{m} p_{ij} . ln(P_{ij}) \tag{8}$$

where $k = \frac{1}{ln(m)}$, $m$ is the total number of farmers, $0 \leq e_j \leq 1$;

Calculate indicator weights:

$$w_j = \frac{1 - e_j}{\sum_{j=1}^{p} (1 - e_j)} \tag{9}$$

### 5.3.3. Calculation of the Composite Score

The endogenous development capacity assessment score was calculated for each farm household:

$$RD_i = \sum_{j=1}^{p} w_j \cdot x'_{ij} \cdot 100 \tag{10}$$

where $e_j$ is the entropy value of j indicators; $w_j$ is the weight of j indicators; $p$ is the total number of indicators; and $RD_i$ is the score of the comprehensive endogenous development capacity assessment of farmers. The weight reflects the degree of contribution of the indicator to the endogenous development capacity of households to rise out of poverty.

This paper adopted the percentage calculation method to determine the weight of each indicator of the performance evaluation system and then obtained the actual value of each indicator through the actual research and the quantitative standards. The actual value of each indicator in each dimension was multiplied by the weight and 100 to obtain the evaluation score of each dimension. The evaluation scores of each dimension were added together to obtain the final score of the capacity assessment. The endogenous development capacity can be observed accordingly. The endogenous development capacity assessment score reflected the endogenous development motivation of the households out of poverty. The higher the value, the stronger the endogenous development ability of the household.

## 6. Empirical Analysis

### 6.1. Analysis of the Impact of IFAD Loan Programs on Farm Household Income and Its Heterogeneity

For villages and farm households with different development levels, there may be differences in the income-increasing effects on farm households generated by the IFAD. Hence, this paper chose three quartiles of 0.25, 0.5, and 0.75 for the quantile regression. The regression results are shown in Table 4.

**Table 4.** Results of quantile regression analysis.

| | Quartile 0.25 | Quartile 0.50 | Quartile 0.75 |
|---|---|---|---|
| Area of agricultural land supported by IFAD | 0.638 *** | 0.522 *** | 0.491 *** |
| | (13.452) | (15.797) | (16.840) |
| Level of knowledge about IFAD | 0.051 ** | 0.076 *** | 0.083 *** |
| | (2.262) | (3.899) | (4.321) |
| Age | 0.041 | 0.010 | 0.029 |
| | (1.559) | (0.432) | (1.052) |
| Educational attainment | −0.010 | −0.013 | −0.016 |
| | (−0.455) | (−0.603) | (−0.651) |
| Physical health status | −0.008 | −0.002 | −0.021 |
| | (−0.362) | (−0.100) | (−0.949) |
| Farming situation | 0.075 ** | 0.074 ** | 0.011 |
| | (2.095) | (2.207) | (0.275) |
| Population burden ratio | 0.452 * | 0.524 *** | −0.079 |
| | (1.872) | (2.684) | (−0.407) |
| Received training or not | 0.025 | 0.073 | −0.054 |
| | (0.330) | (0.958) | (−0.597) |
| $R^2$ | 0.512 | 0.542 | 0.565 |

Notes: * $p < 0.1$ ** $p < 0.05$ *** $p < 0.01$, t-values in parentheses.

According to the regression analysis, the core explanatory variable "area of agricultural land supported by IFAD" has a significant positive effect on the income growth of poverty-stricken households in the "0.25th quintile", "0.50th quintile", and "0.75th quintile". At the same time, the coefficient of 0.638 on lifting low-income households out of poverty in the 0.25 quantile is much greater than that of the 0.50 quantile at 0.522 and the 0.75 quantile at 0.491. Together, the results demonstrate the traits of income from low to high and influence

from strong to weak, indicating that the impact of IFAD support on the income increase in low-income households' poverty relief is marginally decreasing, which coincides with economic rules.

Regarding control variables, the coefficients of age, education level, and physical health status are insignificant from 0, which indicates that age, education level, and physical health status change do not statistically affect the income-increasing effect of farm households. That could be because of the reality that almost all rural farming houses are composed of older individuals. The farming status is positively significant at 0.25 and 0.50 quantile points and negatively significant at 0.75. In low-income and middle-income farming households, the longer spent on farming, the better the crops are, while in higher-income households, there may be a greater tendency to rely on machinery, less time spent on farming, and higher crop quality. The population burden ratio is positively significant at the 0.25 and 0.50 quantile, after which it is statistically insignificant. The sign, which can be positive or negative, is undetermined and may reflect varying household perspectives. Some families may work outside the framing duty due to the family burden, while others choose to stay home and focus on farming for income if they have more dependents to care for closely at home. None of the impact coefficients of whether or not they received training were significant and negative. That may be because 60% of the participants in the IFAD loan program are disadvantaged households in poverty. Therefore, a certain threshold exists for professionals to pass technology and information. The coefficients of the degree of understanding of IFAD are all positive and significant and gradually increase with the quartile. That indicates that the more understanding there is of IFAD, the more farmers choose to join the project, and the more pronounced the effect of income increase. Together, the support of IFAD's loan program has a driving effect on poverty relief households with different income levels, and there is heterogeneity in the income-increasing effect at different income levels. H1 is verified.

*6.2. Analysis of the Impact of IFAD Loan Projects on Farmers' Income Increase Driven by Agricultural Entities*

The results of the weight calculation of the entropy value method are shown in Table 5. The weight value of the loan amount of IFAD is the most significant, 51.722%, indicating that the support of IFAD has the most significant impact on farmers' income increase. Following the ESG score, accounting for 33.992%, the ESG performance can improve the financing attainment and thus promote the increase in farmers' income. In addition, the contribution rate of the control variables accounts for 14.286%, indicating that the context of the Chinese characteristics of a socialist company's financial indicators do not have much impact on farmers' income increase.

**Table 5.** Entropy weighting results.

|  | Information Entropy Value E | Information Utility Value D | Weight (%) |
|---|---|---|---|
| ESG score | 0.937 | 0.063 | 33.992 |
| Controls | 0.973 | 0.027 | 14.286 |
| IFAD loan amount | 0.904 | 0.096 | 51.722 |

The results of the stepwise regression analysis are shown in Table 6. From the analysis of the results of the F test, it can be obtained that the model is highly significant ($p = 0.000$), and the original hypothesis that the regression coefficient is 0 is rejected. The VIF values of the independent variable indicators are all less than 10, indicating that the covariance between the variables has no apparent interaction. From the standardized coefficients, both the IFAD loan amount and ESG score significantly positively impact farmers' income increase. Among them, the coefficient of the IFAD loan amount is more significant. It has a greater impact, which is consistent with results from the previous entropy weighting analysis section. Regarding the ESG score, its coefficient is 0.125 and highly significant

($p$ = 0.000), indicating that the higher the ESG score of agricultural business entities is, the more pronounced the effect of income increase in agricultural households is. H2a and H2b are verified.

**Table 6.** Stepwise regression results.

|  | Standardized Coefficient | t | $p$ | VIF |
|---|---|---|---|---|
| IFAD loan amount | 0.472 | 4.708 | 0.000 *** | 5.737 |
| Total planned investment | 0.443 | 6.947 | 0.000 *** | 2.323 |
| Total actual investment | −0.424 | −5.12 | 0.000 *** | 3.908 |
| Amount reported | 0.231 | 2.696 | 0.007 *** | 4.176 |
| ESG | 0.125 | 2.361 | 0.019 ** | 1.607 |
| $R^2$ | | 0.383 | | |
| Adjusted $R^2$ | | 0.375 | | |
| F | | F = 43.764 ($p$ = 0.000 ***) | | |

Notes: ** $p < 0.05$ *** $p < 0.01$.

*6.3. Results of the Farming Households' Endogenous Development Capacity*

Scholars propose their indicator design programs from different research perspectives and viewpoints for constructing endogenous development capacity indicators. Dong et al. studied six aspects of regional self-development capacity, namely, industrial structure, human resources, financial resources, institutional policies, scientific and technological innovation, and infrastructure [71], while Xu et al. measured the self-development capacity of counties in concentrated contiguous areas using the production capacity of human capital, the capacity for social development, and the carrying capacity of resources and the environment as the second-level indexes, which included seven third-level indexes [72]. Unlike the characteristics of national- and provincial-level statistical yearbook data, this study used the Nanzheng District of Hanzhong City field research data to establish the evaluation index system of endogenous development capacity for village-level poverty relief households. Accordingly, a revised method based on operability, comparability, scientificity, and accessibility was established with the endogenous development capacity of poverty-eradicating households as the first-level index and human capital, natural capital, financial capital, and social capital as the second-level indexes. Human capital was measured via the age of the poverty-eradicating households, their level of education, their physical health, their burden ratio, the number of people working in agriculture in their families, whether or not they have received training, and how much they know about IFAD; natural capital was measured by the area of agricultural land supported by IFAD; financial capital was measured by the net income per capita of the IFAD-supported households; and the presence of village cadres or village representatives in the household measured social capital. The indicator system and the weights of each indicator were calculated by the entropy method, as shown in Table 7.

According to the entropy value method's weight analysis, the degree of education in human capital has the highest weight among driving indicators of the endogenous development capacity of the IFAD–Shaanxi loan project, at 17.946%. As the most important indicator, it reflects the importance of endogenous capabilities such as human ideas and skills. The following weight of the number of family farmers is 17.784%, which mainly reflects the structural composition of family members, and the number of people in the family who choose to work in agriculture has an essential impact on income increase from farming. It reveals the motivation of different households to get out of poverty; it also affects the assessment of the endogenous development capacity. Whether or not they received training accounted for relatively little weight, which could be because village cadres or villagers' representatives actively publicize and encourage poverty relief households participating in IFAD projects to receive training, or only those who participate in training can receive the free distribution of seedlings, fertilizers, and other farming materials, which increases the motivation of poverty-eradicating households. Therefore, the participation of

poverty-stricken households is homogeneous and has less impact on assessing endogenous development capacity.

**Table 7.** Weighting results.

| First-Level Indicators | Second-Level Indicators | Third-Level Indicators | Information Entropy Value E | Information Utility Value D | Weights | Weighting |
|---|---|---|---|---|---|---|
| Endogenous development capacity in poverty-stricken areas | Human capital | Age | 0.937 | 0.063 | 0.1438 | 3 |
| | | Educational attainment | 0.921 | 0.079 | 0.1795 | 1 |
| | | Physical health | 0.979 | 0.021 | 0.0485 | 8 |
| | | Population burden ratio | 0.989 | 0.011 | 0.0259 | 10 |
| | | Number of family farmers | 0.922 | 0.078 | 0.1778 | 2 |
| | | Whether trained or not | 0.984 | 0.016 | 0.0358 | 9 |
| | | Level of knowledge of IFAD | 0.964 | 0.036 | 0.0812 | 5 |
| | Natural capital | Area of agricultural land supported by IFAD | 0.968 | 0.032 | 0.0719 | 6 |
| | Financial capital | Net income per capita | 0.972 | 0.028 | 0.0644 | 7 |
| | Social capital | Whether there are village cadres or village representatives | 0.940 | 0.060 | 0.1353 | 4 |

The comprehensive evaluation score can be acquired accordingly (Table 8), that the six villages as a whole show a medium-level development potential, indicating that the endogenous development capacity of villages in both the sample and the control groups needs to be improved. At the same time, it should be noted that there are apparent differences between villages in the treatment group and the control group with the average composite score of the sample group being 57.391 and the composite score of the control group being 43.852. The above analyses show that in the current situation, the poverty relief people and villages participating in the IFAD loan project show strong endogenous development potential on average. The external financial poverty alleviation support is remarkably effective in increasing the income of poverty-stricken farmers and villages. H3 is verified.

**Table 8.** Comprehensive evaluation.

| Village | Synthesized Assessment | Rankings |
|---|---|---|
| X1 | 43.852 | 6 |
| X2 | 53.377 | 4 |
| X3 | 56.987 | 3 |
| X4 | 58.555 | 2 |
| X5 | 60.643 | 1 |
| X6 | 52.700 | 5 |
| Average composite score of helping villages | 57.391 | |

*6.4. Further Check*

To mitigate the self-selection problem in the modeling setup, referring to Xu et al., this study uses PSM propensity score matching to conduct a counterfactual analysis to derive the role of financial aid from the difference between the control and treated groups [73]. In this paper, farmers participating in the IFAD loan program are set as the treated group, and non-participants are the control group. Variables such as age, education level, physical health status, population burden ratio, number of farmers in the household, whether there

are village cadres or village representatives, and whether they have received training and knowledge of the IFAD loan policy are selected as matching variables. The three matching methods of nearest-neighbor matching within caliper, radius matching, and kernel matching are used to calculate the average treatment effect (ATT), respectively. The results are shown in Table 9. The average treatment effects (ATT) obtained from the three matching methods are all significantly positive at the 1% level, indicating that the IFAD–Shaanxi financial aid policy can promote farmers' income after considering the self-selection bias.

**Table 9.** Results of the propensity score matching.

| Matching Method | ATT | Std. Error | t |
|---|---|---|---|
| Nearest-neighbor matching within the caliper | 0.423 *** | 0.0156 | 27.11 |
| Radius matching | 0.685 *** | 0.0134 | 51.12 |
| Kernel matching | 0.680 *** | 0.0142 | 47.89 |

Note: *** $p < 0.01$.

In this paper, the data were first tested for endogeneity, with a significance *p*-value of 0.000 *** (Table 10), which is statistically significant. Thus, the original hypothesis H0 was rejected, and there was endogeneity in the selected endogenous variables (i.e., ESG and IFAD loans). Further analysis of the data revealed that there may be a self-selection problem in the two variables of ESG and IFAD loans. GMM estimation and 2SLS regression can be used as the solutions to the endogeneity problem. In the presence of heteroskedasticity, GMM is more efficient than that of 2SLS. After examining the data, this study utilized the more suitable GMM estimation. The GMM estimation results show that the Wald value is 390.57, and its significance shows with a *p*-value of 0.000 ***, which rejects the original insignificance hypothesis, indicating that the ESG scores of agricultural entities have a significant positive impact on driving the income of the farm households, and the results are robust.

**Table 10.** GMM estimation results.

| | B | Standardized Coefficient | Z | p | $R^2$ | Adjusted $R^2$ | Wald | Endogeneity Test p |
|---|---|---|---|---|---|---|---|---|
| Amount reported | 0 | −0.21 | −1.065 | 0.287 | | | | |
| ESG score | 0.017 | 1.209 | 11.881 | 0.000 *** | 0.803 | 0.802 | 390.57 | 0.000 *** |
| Amount of IFAD loans | 0 | 0.191 | 1.042 | 0.297 | | | | |

Note: *** $p < 0.01$.

## 7. Conclusions

This study investigates the dual pathways through which IFAD–Shaanxi loan programs influence the increase in farmers' income. The first pathway involves the direct promotion of farmers' income through IFAD loan programs, while the second pathway examines the indirect impact: where these programs drive income increase via agricultural entities. The research employs a quantile regression model to analyze the heterogeneous effects of IFAD loan projects on farmers' income, identifying influencing factors contributing to endogenous motivation for poverty alleviation at different income levels. Additionally, the study evaluates the environmental, social, and governance (ESG) performance of agricultural entities across dimensions such as the environment, society, and corporate governance. It analyzed the impact of IFAD's loan program on farmers' income through agricultural enterprises' ESG performance with entropy weighting and stepwise regression methods. Afterward, from the four dimensions of human capital, natural capital, financial capital, and social capital, this study constructed the evaluation index system of IFAD's

endogenous development capacity for empowering poverty-eradicating households, assessed the endogenous development capacity of poverty-eradicating farmers participating in the loan program, and put forward countermeasure suggestions for sustainable poverty alleviation for farmers at different income levels to promote the sustainability of farmers' income increase.

The research conclusions can be summarized as follows: (1) IFAD loan projects had a significant positive effect on the income increase in farm households. The heterogeneity analysis showed that IFAD loan projects had different impacts on the income increase in farm households at different income levels, and the lower the income level, the more pronounced the promotion effect [7,33,47,74]. (2) IFAD loan projects drove the income increase in farm households through agricultural entities, in which the mechanism of the channel was its ESG performance. (3) The poverty relief people and villages participating in IFAD loan projects show strong endogenous development dynamics on average [9,75].

The findings have added value to the related literature. First, it broadens the literature on more detailed studies of financial policies to promote farm household income; second, it expands the literature on the combination of ESG performance of agricultural business entities and farm household income and provides insights into China's rural revitalization and sustainable development. This study is more applicable to other countries that emphasize rural issues and value the interests of farmers. In addition, the findings contribute to the debate about the significance of specific financial aid programs concerning low-income farmers in developing countries. They demonstrate evidence that financial aid matters. More specifically, in China's dualistic social structure, this work examines socioeconomically disadvantaged groups and focuses on their learning effects in trickle-down theory. First, it is essential to encourage villages to actively participate in financial aid programs matching their endowments, such as abundant resources. Second, countries should pay more attention to the matches and scope of the application of financial aid as well as their effectiveness [76].

This study has strong external validity and can be generalized to countries with similar endowments and values. Firstly, in terms of finding representativeness, this study can be generalized to countries with endowments similar to China's. For example, Ghana's agriculture is practiced primarily on a small-scale basis, with most farms (76 percent) operating on less than 2 ha of land and located in rural areas (FAO, 2016) [14]. It is similar to Chinese agriculture and has possible research potential. Alternatively, the finding could give insight into countries with shared prosperity and following a sustainable development path. Secondly, regarding values, this study is more applicable to countries that emphasize rural issues and value the interests of farmers. At the same time, limitations should be addressed in future research. This paper could have data self-selection issues. All twenty-five villages participating in IFAD–Shaanxi loan programs have specialty industries, and 20% were selected and researched. Those five villages with specific industries could make the case that they have a better foundation for developing specialty industries, and subjectively, these villages' cadres have more sophisticated leadership skills and better social networks. Also, possible construction indicators issues, especially ESG indicators, could be improved. China is paying more attention to these topics. However, the development of these indicators is at an early stage in China's rural revitalization practice. Establishing those indices should be a dynamic process combining China's development practice and specific realization cases.

## 8. Policy Implication

Policymakers ought to increase policy publicity to enhance the poor households' awareness of self-poverty eradication. The publicity of the policy is insufficient, and the interviews revealed that information asymmetry is the most critical issue to be emphasized, as some respondents were not fully aware of the IFAD loan policy. Information asymmetry may be reflected in poor households' low awareness of self-degradation and low willingness to eliminate poverty. Given that financial support can stimulate the endogenous

development of poverty-stricken households and poverty-stricken areas, it is necessary to increase the publicity of IFAD and similar policies to enhance the motivation of farmers to lift themselves out of poverty, prompting them to take the initiative to participate in financial poverty alleviation programs, which in turn will lead to more poverty-free people [6].

Another recommendation is to increase training for poor households and stimulate their endogenous motivation. According to the surveys, the professional farmer group had significant internal power to enhance farming revenue and frequently acted as an example when implementing ideas. Therefore, it is essential to consistently raise the general standard of living in low-income households, expand education to fight poverty, develop infrastructure, and halt the passing down of poverty from one generation to the next. It is vital to utilize financial policy support to help impoverished people improve their thinking [74]. That could contribute to ensuring the sustainability of the effects of financial support, strengthening the income-generating capability of poor households, accomplishing the objective of sustainable development, consolidating the results of poverty alleviation, and improving the quality of poverty alleviation.

Continuing the policy of financial poverty alleviation, the "blood-forming" approach to poverty alleviation has been a regular feature of rural revitalization. The endogenous development motivation of those lifted out of poverty was mainly influenced by human capital with the most significant effects being the number of people working in agriculture and the average number of years of schooling in the household. It is necessary to promote the enrollment of school-age children of poverty-stricken households/regions and apply for education subsidies to ensure that school-age children of poverty-stricken households are enrolled in school and gradually improve the literacy level of poverty-stricken households. At the same time, it is crucial to boost support for regional specialty sectors, upgrade the industrial landscape, and encourage agricultural modernization. In addition, encouraging talent to return to their hometowns is needed to promote the revitalization of talent, the comprehensive development of the rural, and the sustainable development of poverty alleviation [7].

Lastly, it is necessary to actively construct ESG evaluation standards with Chinese characteristics to enhance corporate social responsibility. Sustainable development is the central theme of the times, and ESG performance is regarded as a core indicator of corporate sustainability. Aligning with China's commitment to achieving the goals of "carbon peak" and "carbon neutrality", ESG is an essential tool for implementing sustainable development goals. Adhering to and developing socialism with Chinese characteristics, Chinese agricultural enterprises are responsible for developing the economy, improving people's livelihoods, and revitalizing the countryside. However, it does not align with China's national conditions and agricultural situation to copy Western enterprises' ESG theories and practices. Thus, establishing widely applicable, comprehensive, and quantifiable corporate ESG indicators, assessment systems, and regulatory mechanisms for Chinese agribusinesses is necessary in light of China's reality [47].

**Author Contributions:** Conceptualization, H.P., L.Y., C.Z., Y.Z. and Y.G.; Methodology, H.P., L.Y., C.Z. and Y.Z.; Investigation, H.P., L.Y., C.Z., Y.Z. and Y.G.; Formal analysis, H.P., L.Y., Y.Z. and Y.G.; Resources, H.P. and L.Y.; Writing—original draft preparation, H.P., L.Y., C.Z., Y.Z. and Y.G.; Writing—review and editing, H.P., L.Y., Y.Z. and Y.G.; Visualization, H.P. and Y.Z.; Supervision, L.Y.; Project administration, H.P. and L.Y. All authors have read and agreed to the published version of the manuscript.

**Funding:** This research was funded by Shaanxi Natural Science Research Funding (grant number: 2023-JC-QN-0801), Northwest A&F University Research Fund (grant number: Z1090122041), and Shaanxi Postdoctoral Research Fund (grant number: 2023BSHEDZZ129). The APC was funded by the Northwest A&F University Research Fund (grant number: Z1090122041).

**Institutional Review Board Statement:** Not applicable.

**Informed Consent Statement:** Not applicable.

**Data Availability Statement:** The datasets used and/or analyzed during the current study are available from the corresponding author on reasonable request.

**Acknowledgments:** The authors highly appreciate the anonymous reviewers and the editors for their constructive comments and suggestions that helped us to improve the paper.

**Conflicts of Interest:** The authors declare no conflicts of interest.

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
