# Peer review of "Research on Financial Poverty Alleviation Aid for Increasing the Incomes of Low-Income Chinese Farmers"

_sustainability, doi:10.3390/su16031057_

Round 1
Reviewer 1 Report
Comments and Suggestions for Authors
The article raises important practical issues related to the use of fiscal policy instruments (mainly taxation) in the process of sustainable income development of various social groups in China (example of farmers).
The research methodology was correctly applied and the research was conducted with a multidisciplinary character and taking into account the assumptions of the Chinese government's mnikroeconoic polutics.
Induction and synthetic and analytical methods of data analysis were used.
The study lacks consideration of the social and socio-socioeconomic issues of farmers from the perspective of the effectiveness of classical fiscal intervention instruments.
How religion, traditions and level of financial literacy affect the effectiveness of policies to support the level of income development? (financial lliteracy)
Author Response
On behalf of all contributing authors, I sincerely appreciate your constructive comments concerning our paper. These comments are all valuable and helpful for improving our manuscript. According to your comments, we have extensively modified our manuscript and provided extra references to make our research findings more convincing. We hope those changes can relieve your concerns.

Reviewer 2 Report
Comments and Suggestions for Authors
This paper examines the impact of the International Fund for Agricultural Development (IFAD) loan project on increasing the income of low-income farmers in China. Using survey data from households in villages participating in the IFAD project, the authors employ quantile regression and stepwise regression models to analyze the direct and indirect effects of IFAD loans on farmers' income growth. They also construct an index to evaluate the endogenous development capacity of IFAD-supported households.
The topic is highly relevant given China's focus on poverty alleviation and rural development. The paper makes a worthwhile contribution by investigating the heterogeneous effects of IFAD loans across different income groups and the mechanisms through which agricultural enterprises' ESG performance affects income growth. The empirical analysis is reasonably rigorous.
(i) Key Strengths:
-Examines an important policy issue of how financial aid affects rural poverty alleviation in China, which aligns with the scope of the journal.
-Employs appropriate econometric methods (quantile regression) to test research hypotheses related to heterogeneous treatment effects.
-Incorporates mediation analysis and ESG factors into the stepwise regression model to shed light on indirect effects and mechanisms.
-Constructs an endogenous development capacity index for poverty-alleviated households using multiple indicators.
-Uses primary data from household surveys and enterprise interviews to complement available secondary data.
(ii) Primary Weaknesses:
-The literature review is quite minimal and lacks reference to similar studies on the impact of financial inclusion and agricultural aid programs in other developing country contexts.
-The sampling methodology for the household survey is not sufficiently described. Details on sampling technique and selection of villages should be provided.
-Potential issues of endogeneity/self-selection in program placement and participation are not addressed.
-The indices constructed for ESG performance and endogenous development capacity could benefit from more justification of selection of indicators and weighting.
(iii) Modification Suggestions:
-Expand the literature review to synthesize relevant studies on the research topic and situate the paper within the broader scholarship.
-Provide more details on the sampling methodology, sample size, and representativeness of the household survey data.
-Provide limitations related to endogeneity concerns in the empirical analysis and consider additional robustness checks.
-Include more discussion on the construction of indices - choice of indicators, weighting, sensitivity analysis, etc. In addition, the Discussion section needs to be a coherent and cohesive set of arguments that take us beyond this study in particular and help us see the relevance of what the authors have proposed. The authors need to contextualize the findings in the literature and need to be explicit about the added value of your study towards that literature. Also, other studies should be cited to increase the theoretical background of each of the methods used. Findings should be contextualized in the literature and should be explicit about the added value of the study towards the literature. The contribution and implications of the article are yet to be specified. Please refer to the style, DOI: 10.1177/2158244020902088
-Proofread the paper carefully to fix minor grammar errors and typos.
Comments on the Quality of English LanguagePolish the paper carefully to fix minor grammar errors and typos.
Author Response

(The authors gave the same response as above.)

Reviewer 3 Report
Comments and Suggestions for Authors
Thank you for the opportunity to review the manuscript "Research of the Impact of Financial Poverty Alleviation on the Income-Increasing Path of Chinese Low-Income Farmers". The manuscript addresses a relevant and important topic for Chinese low-income farmers. There are some points to be improved to increase the impact and audience of the research.
Introduction. The first two paragraphs of the manuscript present the context of the research. The text is clear and well-written but could be shortened a little.
Literature review. The first paragraph of the literature review presents the justifications for the research while informing what theoretical frameworks are used in the research. The following two paragraphs present the theoretical frameworks. The fourth paragraph is quite obscure. There is an attempt to present the relevance of combining theoretical frameworks as something innovative and useful for understanding the problem. The aim of combining theoretical frameworks and employing the indicated statistical techniques is positive. However, the focus of this paper is not methodology. Therefore, it is necessary to make the objective of the manuscript clearer. It is essential to clarify in the text what problem the research seeks to overcome, the research gap, and, above all, what the objective is and the impact expected when achieving it.
Furthermore, these elements must be in the introduction. A sentence summarizing each supporting literature should be created, associated with the fourth paragraph (reformulated), and moved to the introduction. Finally, clarifying the research results' relevance and applicability to a wider audience is important. How do the results apply to other countries?
Theoretical analysis and research hypotheses. Section 3 is presented in detail. The hypotheses are well-founded and formulated. In practice, section 3 is an extension of the literature review section, and it is desirable to merge them after the correction suggested for the introduction. The remaining parts of the manuscript are well-written. However, the text is somewhat prolix. Making the text more objective and shorter is desirable to increase readability and readers' interest in the paper.
It would be very interesting to see an analysis using structural equation modeling in future research [1]. Many questions could be answered. Which dimensions are internally consistent? Which sub-indicators should be discarded? How do the dimensions relate? What are the weights of the sub-indicators and dimensions generated by this model? Etc.
· • Measuring intra-urban inequality with structural equation modeling: A theory-grounded indicator
Minor errors
Notation (1): 𝑙𝑛𝑌 or L𝑛𝑌? 𝜀 and k were not defined.
Notation (2): 𝐼𝐷, 𝐿, and 𝑡 were defined in (4).
Please check that the notation terms are all presented in the correct location.
Author Response

(The authors gave the same response as above.)

Reviewer 4 Report
Comments and Suggestions for Authors
Excellent study that can be replicated in other countries. Excellent application of ESG for developing country standards that can be used for global comparisons.
Author Response

(The authors gave the same response as above.)

Reviewer 5 Report
Comments and Suggestions for Authors
Thanks for the opportunity to review the article, I find it interesting one.
1. The abstract is well written and it describes the whole story of the article.
2. There is enough literature review and previous studies have been discussed deeply.
3. Section 3.1 needs more references as it is not possible whole section is based on single reference.
4. Please shorten the developed hypotheses, they are too long.
5. Kindly build your study on a theory as it will strengthen the study and support its grounding.
6. Please inform readers that this is a quantitative deductive study and focus more on the research methodology.
7. Change the word “variables definitions” to “concepts definition”.
8. Please don’t mix research methodology and data analysis as these are two separate sections.
9. Provide a separate section for the theoretical implications of the study and separate implications from the conclusion.
10. Please proofread the article, there are many places where mistakes are there.
Thanks
Comments on the Quality of English LanguageMinor
Author Response

(The authors gave the same response as above.)

Round 2
Reviewer 2 Report
Comments and Suggestions for Authors
The authors have made a great effort to address the issues raised by referees. The work done has resulted in an paper that can be published as is.